# *Citrus junos* Tanaka Peel Extract and Its Bioactive Naringin Reduce Fine Dust-Induced Respiratory Injury Markers in BALB/c Male Mice

**DOI:** 10.3390/nu14051101

**Published:** 2022-03-05

**Authors:** Dong-Hun Lee, Jin-Kyung Woo, Wan Heo, Wen-Yan Huang, Yunsik Kim, Soohak Chung, Gyeong-Hweon Lee, Jae-Woong Park, Bok-Kyung Han, Eui-Chul Shin, Jeong-Hoon Pan, Jae-Kyeom Kim, Young-Jun Kim

**Affiliations:** 1Department of Food and Biotechnology, Korea University, Sejong 30019, Korea; dhanmin@korea.ac.kr (D.-H.L.); wlsrud5877@korea.ac.kr (J.-K.W.); flyhighwy@korea.ac.kr (W.-Y.H.); hanmoo@korea.ac.kr (B.-K.H.); 2Department of Food Science and Engineering, Seowon University, Cheongju 28647, Korea; 01062033526@seowon.ac.kr; 3Lotte R&D Center, Seoul 07594, Korea; yunsik.kim73@lotte.net (Y.K.); soohak.chung@lotte.net (S.C.); ghlee@lotte.net (G.-H.L.); jaewoong_park@lotte.net (J.-W.P.); 4Department of Food Science, Gyeongsang National University, Jinju 52828, Korea; eshin@gnu.ac.kr; 5Department of Behavioral Health and Nutrition, University of Delaware, Newark, DE 19716, USA; jhpan@udel.edu (J.-H.P.); jkkim@udel.edu (J.-K.K.)

**Keywords:** *Citrus junos* Tanaka, naringin, fine dust, particulate matter 10, pulmonary, apoptosis

## Abstract

Particulate matter (PM) 10 refers to fine dust with a diameter of less than 10 µm and induces apoptosis and inflammatory responses through oxidative stress. *Citrus junos* Tanaka is a citrus fruit and contains bioactive flavonoids including naringin. In the present study, we aimed to identify the preventive effect of *Citrus junos* Tanaka peel extract (CPE) against PM_10_-induced lung injury. As a proof of concept, NCI-H460 cells were treated with CPE (800 μg/mL, 12 h) in conjunction with PM_10_ to examine intracellular antioxidative capacity in the pulmonary system. In an in vivo model, male BALB/c mice (*n* = 8/group) were randomly assigned into five groups: NEG (saline-treated), POS (PM_10_ only), NAR (PM_10_ + naringin, 100 mg/kg), CPL (PM_10_ + CPE low, 100 mg/kg), and CPH (PM_10_ + CPE high, 400 mg/kg). Intervention groups received dietary supplementations for 7 days followed by PM_10_ exposure (100 mg/kg, intranasal instillation). Compared to the NEG, the CPE decreased to 22% of the ROS generation and significantly increased cell viability in vitro. The histological assessments confirmed that pulmonary damages were alleviated in the PM_10_ + CPL group compared to the POS. Pro-inflammatory cytokines and NF-κB/apoptosis signaling-related markers were decreased in the PM_10_ + CPL group compared to the POS. These results indicated that CPE showed promising efficacy in preventing pulmonary injuries in vivo. Such protection can be explained by the anti-oxidative capacity of CPE, likely due to its bioactives, including naringin (7.74 mg/g CPE). Follow-up human intervention, as well as population-level studies, will further shed light on the preventive efficacy of CPE against pulmonary damage in humans.

## 1. Introduction

Particulate matter 10 (PM_10_) refers to fine dust with a diameter less than 10 µm, and it is known that the smaller the particle size, the more fatal [1]. Exposure to PM_10_ can cause bronchial and cardiovascular diseases. Since it is mainly absorbed through respiration, many studies related to pulmonary inflammation were reported, while reactive oxygen species (ROS)/apoptosis/NF-κB are spotted as major mechanisms to explain its toxicity [2]. Thus, it is a reasonable premise to identify a safe dietary antioxidant(s) to mitigate PM_10_-induced pulmonary damages. Relatedly, studies demonstrated that natural product extracts (e.g., tart cherry and pomegranate) reduce inflammatory response induced by PM_10_ in various study models [3,4,5]. For instance, Lee et al. reported that medicinal herb extract ameliorates PM_10_-induced inflammatory responses; responsible active compounds therein were antioxidants (e.g., baicalin and schizandrin) [6]. Hitherto, few human studies are available if chronic dietary antioxidants (and/or a mixture of extracts) are negatively associated with PM_10_-related diseases.

*Citrus junos* Tanaka (also known as Yuja or yuzu) is a citrus fruit of the family *Rutaceae* and is mainly consumed as tea and sauce [7]. It has been long used as a traditional medicine in Northeast Asian countries to treat bronchial and respiratory systems [8]. In particular, in its peel, there are various bioactives including naringin and hesperidin; naringin is a well-known active compound that can attenuate inflammation, cytotoxicity, and oxidative stress [9]. Preventive potential yuzu was reported in various disease models; one example is a preclinical study where *Citrus junos* Tanaka prevented dextran sodium sulfate-induced colitis via inhibiting the activation of endogenous NF-κB signaling pathway by suppressing the translocation of p65 [10]. In another study, *Citrus junos* Tanaka and its bioactive (i.e., naringin) ameliorated pulmonary damages in acrolein inhaled mice models [11,12]. *Citrus junos* Tanaka decreased ROS accumulation and p53-dependent apoptotic signaling and naringin suppressed phosphorylation of p65 via quenching acrolein-induced ROS. Studies utilized acrolein-induced pulmonary provide mechanistic insights related to yuzu and naringin, yet an evitable limitation is that humans are exposed to a mixture of inhalation toxicants (e.g., PM_10_), thus making it less translational.

Despite the wide scope of physiological effects of *Citrus junos* Tanaka and its bioactive compound (naringin), no research has examined their roles in PM_10_-induced respiratory toxicity. To this end, our study aimed to test our hypothesis that water extract of *Citrus junos* Tanaka and naringin will prevent PM_10_-induced pulmonary inflammatory markers and lung epithelial cell death.

## 2. Materials and Methods

### 2.1. Materials and Reagents

The *Citrus junos* Tanaka peel used in this study was obtained from the Goheung Agricultural Cooperative Federation (Goheung, Korea), and the *Citrus junos* Tanaka peel extract (specimen voucher number: GFS-#003) used in this study was verified and stored in the specimen room located in the Department of Food Science of Gyeongsang National University. It was ground using a blender and extracted with 10 volumes of hot distilled water (60 °C to 100 °C) for 24 h. The extract (hereafter CPE) was filtered through a filter paper (Whatman; Maidstone, UK), freeze-dried, and stored in a −80 °C deep-freezer. Naringin (≥95%) and PM_10_ (ERM-CZ120) used for certified reference material were purchased from Sigma (St. Louis, MO, USA).

### 2.2. Analysis of Naringin Contents

The Agilent 1260 series (Agilent Technologies; Santa Clara, CA, USA) that was equipped with a quaternary pump, an online degasser, an auto plate-sampler, and a photodiode array detector with an Agilent ZORBAX Eclipse XDB-C18 column (4.6 mm × 150 mm, 5 µm particle size) was used for the separation. Solvent A consisted of 0.1% formic acid in water and solvent B consisted of 0.1% formic acid in acetonitrile. The solvent gradient was listed: 10% B at 0–5 min; 10–20% B at 5–13 min; 20% B at 13–20 min; 20–50% B at 20–25 min; 50–80% B at 25–30 min. The composition was next held at 95% B for 5 min, and then it returned to its initial conditions and was maintained for 5 min to equilibrate the column. The detection wavelength was 280 nm. The column temperature was maintained at 28 °C with a thermostatically controlled column compartment. The flow rate was 1.0 mL/min, and the injection volume was 10 μL.

### 2.3. Cell Culture

The human lung carcinoma cell line NCI-H460 was obtained from the Korean cell line bank (Seoul, Korea). Cells were cultured in RPMI-1640 (Thermo-fisher Scientific, Waltham, MA, USA) supplemented with 10% fetal bovine serum, 1% penicillin–streptomycin, and 0.2% NaHCO_3_, and grown in a 5% CO_2_ incubator at 37 °C.

### 2.4. Intracellular Anti-Oxidative Assay

NCI-H460 cells were seeded at a density of 3 × 10^4^ cells per well on a 96-well plate. After 24 h, the growth medium was removed, and the wells were washed with phosphate-buffered saline solution. Then, 100 μL of the samples (800 μg/mL, both NAR and CPE) were treated on the cell with 25 μM of dichloro-dihydro-fluorescein diacetate. After an hour, the wells were washed with phosphate-buffered saline solution again and 600 μM of 2,2′-azobis(2-methylpropionamidine) dihydrochloride (AAPH) or 0.4 mg/mL of PM_10_ was added to the cells in 100 μL of Hank’s balanced sale solution. The plate was placed into a fluorescence plate-reader and measured with 538/485 nm every 5 min for an hour. The negative control wells contained cells treated with dichloro-dihydro-fluorescein diacetate and Hank’s balanced sale solution without AAPH or PM_10_. Assay results were expressed as a percent of control.

### 2.5. Cell Viability Assay

NCI-H460 cells were plated onto a 96-well plate at a density of 3 × 10^4^ cells per well and exposed to naringin or CPE (800 μg/mL) and PM_10_ (400 μg/mL) for 12 h. Thereafter, the 3-(4,5-dimethylthiazol-2-yl)-2,5-diphenyltetrazolium bromide reagent solution was added to each well at 37 °C. After two hours, the formazan dissolved with dimethyl sulfoxide (100 μL), and absorbance was measured at 595 nm using a microplate reader.

### 2.6. Animal and Treatment

Male 7-week-old BALB/c mice were obtained from the Raonbio (Yongin, Korea). The animals were housed in a controlled environment with 12 h light and dark cycles. The mice received water and a standard diet ad libitum. All mice were randomly divided into five groups (*n* = 8) as follows: (1) NEG (saline + vehicle), (2) POS (saline + PM_10_), (3) NAR (naringin 100 mg/kg + PM_10_), (4) CPL (CPE 100 mg/kg + PM_10_), (5) CPH (CPE 400 mg/kg + PM_10_). Naringin and CPE were treated orally with saline once daily for 7 days, and 100 mg/kg of PM_10_ was treated by intranasal administration only once on the last day. After 6 h of PM_10_ treatment, all groups were sacrificed using an overdose of avertin. The lung and bronchoalveolar lavage fluid (BALF) was collected for immunoblot analysis, histochemical analysis, and enzyme-linked immunosorbent assays (ELISAs).

### 2.7. Histological Analysis

The isolated lung tissues were fixed in 10% formalin, dehydrated using a graded series of alcohol, and embedded in paraffin. The paraffin molds were sectioned with 3 μm. Thereafter, they were deparaffinized, rehydrated with alcohol, and stained with hematoxylin and eosin.

### 2.8. Immunoblot Analysis

Lung tissues were homogenized in lysis buffer with protease inhibitor and phenylmethylsulfonyl fluoride to prepare protein lysates. Proteins were resolved by 10% sodium dodecyl sulfate-polyacrylamide gel electrophoresis and transferred to nitrocellulose membranes. The membranes were blocked with 3% bovine serum albumin solution for 1 h at room temperature. Then, the membranes were probed with primary antibodies overnight and incubated with peroxidase-conjugated secondary antibodies. The protein bands were visualized using enhanced chemiluminescence reagent on an ImageQuant LAS-4000 imager (General Electric, Pittsburgh, PA, USA) and quantified using ImageJ software (National Institute of Health, Bethesda, MD, USA).

### 2.9. ELISA Assay

BALF was collected and stored at −80 °C. Tumor necrosis factor-α (TNF-α) and interleukin-1β (IL-1β) levels in BALF were measured using mouse-specific ELISA kits according to manufacturer’s instructions (Abcam; Cambridge, UK).

### 2.10. Statistical Analysis

Data were analyzed by *t*-test and one-way analysis of variance using SAS software (SAS Institute, Cary, NC, USA). The least-squares mean option using a Tukey–Kramer adjustment was used for multiple comparisons among the experimental groups. Data are shown as the means ± SEM (standard error of the mean). *p*-values of <0.05 were considered statistically significant.

## 3. Results and Discussion

### 3.1. Naringin Content Was Highest in the 60 °C Water Extract of Citrus junos Tanaka Peel

The peel of *Citrus junos* Tanaka, which is rich in active compounds such as phenolic compounds (e.g., limonene, rutin, naringin, narirutin, hesperidin, and neohesperidin), is mainly used as a dried product for herbal medicine, tea, or beverages [7,13,14,15,16]. The various functions of these compounds have been reported such as immune and inflammation regulation [14,17,18,19,20,21]. In particular, naringin has been reported in several studies, showing a key protective effect against oxidative stress [17,22]. To identify optimal extraction temperature, *Citrus junos* Tanaka peel was extracted using water as a solvent under conditions of 60 °C, 80 °C, and 100 °C; these temperatures were selected to reflect its daily use. After this, the contents of naringin in each fraction were subjected to the high-performance liquid chromatography system (see detailed analytical conditions in the Materials and Methods section) with a commercial reference standard for quantification. We noticed that naringin contents tended to decrease with higher temperature, although it was not greatly impacted. Among the conditions, CPE prepared at 60 °C presented the highest naringin content, and thus it was selected for further in vitro and in vivo analyses (7.74 mg/g naringin; Figure 1).

### 3.2. CPE Treatment Protected Cells from PM_10_-Induced Oxidative Stress

PM_10_ provokes oxidative stress to the keratinocyte, bronchial, and alveolar epithelial cells due to its direct exposure [23,24]. Before checking the protective activity, we executed cytotoxicity tests using the MTT assay to determine the best concentration of naringin and CPE (Appendix A). The protective potential of CPE was first examined, as a proof of concept, using in vitro intracellular anti-oxidative assay; the AAPH was used as a positive control in the model. Both AAPH and PM_10_ treatment significantly increased intracellular ROS levels compared to the NEG group (1000% and 1450%, respectively; *p* < 0.05; Figure 2A), and quercetin, a reference antioxidant, reduced ROS generation successfully (90% compared to POS group; *p* < 0.05; Figure 2B), guaranteeing that the experimental model works well. As expected, the groups treated with naringin and CPE showed a dramatic reduction in ROS generation in the model (35% and 23%, respectively; *p* < 0.05; Figure 2B). Oxidative stress can cause apoptotic cell death, which is an initial response to the development of respiratory manifestations [25,26]. We confirmed that cell viability was significantly decreased in the PM_10_-treated group (28% compared to the NEG group; *p* < 0.05), while CPE- and naringin treated groups showed suppressed cell death compared to the PM_10_-treated group (>90% compared to the NEG group; *p* < 0.05 compared to POS group; Figure 2C). With the potent anti-oxidative capacity and protection against PM_10_-induced cell deaths, the in vitro study results led us to further verify physiological effects in a whole-body animal model system.

### 3.3. CPE Alleviated PM_10_-Induced Pneumonitis via Suppression of Pro-Inflammatory Cytokines

To determine whether the administration of CPE and naringin could alleviate PM_10_-induced lung inflammation, mice were subjected to oral administration of CPE and naringin for 7 days; after this, to induce pneumonitis phenotypes, the mice were subjected to intranasal injection of PM_10_ (100 mg/kg body weight) at the last day and sacrificed after 6 h. The naringin and low-dose CPE concentration (i.e., 100 mg/kg body weight) were determined on the basis of the work of previous papers using similar samples (i.e., naringin and natural product extracts), and the higher dose of CPE was chosen to confirm the dose-dependent response of CPE [11,27,28,29,30,31,32,33]. As an initial response against PM_10_, inflammatory immune cells (e.g., eosinophils and neutrophils) are recruited on the site, followed by air space enlargement and detachment, all of which are typical pneumonitis manifestations [34,35]. 

To determine if our PM_10_ treatment conditions induce pneumonitis phenotypes in mice, histological staining was performed. As presented in Figure 3, lung pathology assessment shows that bronchoalveolar structure was destructed by PM_10_ treatment, while, qualitatively, sample treatment groups showed marginal improvements; specifically, pulmonary alveolar size was increased in the POS group (457% compared to NEG group; *p* = 0.0572; Figure 3B), whereas the NAR, CPL, and CPH presented protection, albeit statistical significance was not reached (54%, 58%, and 49% compared to POS group, respectively; *p* = 0.1224, *p* = 0.0915, *p* = 0.1737, respectively; Figure 3B).

To understand preventive mechanisms of CPE at the molecular level, pro-inflammatory cytokines, TNF-α and IL-1β, in BALF were first analyzed. In agreement with our histological observations, the pro-inflammatory cytokines were both dramatically increased by PM_10_ treatment (750% for TNF-α and 144% for IL-1β compared to NEG group, respectively; *p* < 0.05; Figure 4), while naringin and a low dose of CPE decreased TNF-α (65% and 38% for naringin and CPL, compared to the POS group, respectively; *p* < 0.05) and IL-1β (≈31% for both naringin and CPL, compared to the POS group, respectively; *p* < 0.05; Figure 4). TNF-α is a pro-inflammatory cytokine and is secreted by immunocytes such as macrophages, monocytes, and neutrophils; it responds to inflammation or external stress factors to transactivate NF-κB and its downstream target genes [25]. Researchers have shown that TNF-α mediates the recruitment of neutrophils and eosinophils during airway inflammation [36]. On the other hand, epithelial cells were damaged, and danger signals (e.g., uric acid and adenosine triphosphate) activated toll-like receptor 4 and inflammasome (e.g., NOD-like receptor porin domain-containing protein 3). NF-κB is translocated to the nucleus through the activated toll-like receptor 4 signaling pathway, thereby pro-IL-1β is a transcript and is activated as IL-1β by caspase 1 provokes a neutrophilic and eosinophilic inflammatory response [37]. Consequently, the increase of TNF-α and IL-1β in BALF indicated that intranasal treatment of PM_10_ provokes pneumonitis, while CPE and naringin treatment counteracts against NF-κB-mediated inflammatory responses due to decreased TNF-α and IL-1β. However, at this moment, it is unclear as to why a high dose of CPE (i.e., CPH in Figure 4) failed to recapitulate the protection. Further pharmacokinetic, as well as bioavailability studies, are warranted.

### 3.4. CPE Downregulated NF-κB and Caspase Cascade Signaling Pathways in Lung Tissue

Overproduction of ROS in the body causes inflammatory responses, which could lead cells to apoptosis, and inhaled PM_10_ is a well-established inflammation mediator by generating ROS in the lungs [38,39]. To determine whether intranasally administered PM_10_ induces a local inflammatory response in this model, a representative downstream cytokine of NF-κB, IL-6, was assessed using lung tissues. The expression level of IL-6 was significantly increased by PM_10_ treatment (158% compared to NEG group; *p* < 0.05; Figure 5A), whereas it was significantly decreased in CPE and naringin administration groups (37% and 29% compared to POS group; *p* < 0.05; Figure 5A). Relatedly, the mechanism of apoptosis can be divided into two broad categories: intrinsic and extrinsic pathways [40]. When cells recognize DNA damage, mitochondria release cytochrome c, which activates caspase-9 and caspase-3 in a cascade, leading to apoptosis in the intrinsic pathway. In the extrinsic pathway, external stimuli such as ROS and/or inflammatory cytokines activate caspase-8 and subsequently caspase-3, thereby leading to apoptosis [41]. We explored if inhaled PM_10_, as an external stimulus, triggers the extrinsic apoptosis pathway, secreted Fas, and its downstream apoptotic markers (i.e., c-caspase-8 and c-caspase-3) that were quantified in the lung tissues.

Fas, c-caspase-8, and c-caspase-3 were significantly increased by PM_10_ treatment (Fas, 209%; c-caspase-8, 156%; c-caspase-3, 202%, compared to the NEG group; *p* < 0.05; Figure 5), while naringin reversed them all (Fas, 60%; c-caspase-8, 44%; c-caspase-3, 75%, compared to the POS group; *p* < 0.05; Figure 5). Likewise, the CPL had similar protection in c-caspase-8 and c-caspase-3 (43% and 49%, respectively; *p* < 0.05; Figure 5C,D), whereas the CPH only reduced c-caspase-3 (50% *p* < 0.05; Figure 5D). Our observations are in good agreement with others. In a previous study, CPE protected acrolein-induced lung damage by significantly reducing c-caspase-3 and c-parp [11]. Likewise, the caspase cascade and pro-inflammatory reactions are key proteins in the context of airway toxicant-induced pulmonary injury models (e.g., PM) [42,43,44,45,46,47]. Hence, results herein, in addition to evidence in the literature, collectively indicate that targeting extrinsic apoptosis by mitigating ROS production would be an effective approach to mitigate inflammatory responses. Our in vitro data confirm that naringin would be, at least in part, responsible for such production.

A few strengths of the study should be noted. First, we established the contents of naringin in CPE through high-performance liquid chromatography, which allows us to estimate the exact level of exposure of reference constituent from the CPE. Second, we established in vivo pneumonitis via nasal administration, which is more technically challenging yet relevant to reflect inhalation toxicant exposure. To examine dose-dependent responses of CPE, we included both low and high doses of CPE; as a result, we were not able to demonstrate further protection from the high-dose group (i.e., CPH), which is fundamental information for future pharmacokinetic and toxicity studies. In contrast, weak points also existed. We analyzed the content of naringin, which is expected to be a major bioactive of CPE, through high-performance liquid chromatography, but other miscellaneous phenolic compounds were not quantified for this study. Therefore, it is certainly possible that other potential phytonutrients may have played roles that can be further investigated in future studies. Relatedly, we have shown that naringin inhibits ROS generation in vitro, in which its potential was compared with a known radical scavenging compound (i.e., quercetin). Thus, the inclusion of a positive control group (using a known ROS scavenger) might have provided additional insight as to relative potential against PM_10_-induced respiration damage. Although it was a proof-of-concept study, we used human lung carcinoma, which is not relevant to the prevention study of pneumonitis; perhaps primary normal alveolar epithelial cells would have been a better option. However, our aim to use the in vitro model was to simply examine the intracellular anti-oxidative potential of naringin and CPE before the in vivo study in which the cell study results were reproduced. Second, our study conditions were acute, and hence were not harsh enough to induce further advanced pathological markers (e.g., pulmonary fibrosis); thus, long-term exposure studies might be warranted in the future.

## 4. Conclusions

This study provides evidence that ROS production in lung cells increased by PM_10_ treatment can induce oxidative damage-induced extrinsic apoptosis, which could affect cell viability. We confirmed that CPE and naringin treatment decreased intracellular ROS production and increased cell viability of lung cells against PM_10_-induced oxidative stress. Furthermore, we investigated the protective effect of CPE against oxidative stress by a PM_10_-induced lung injury mouse model and confirmed that it could alleviate the damage through histological complementation and suppressed the expression of pro-inflammatory cytokines and extrinsic apoptosis-related proteins. In conclusion, these results indicate that CPE exhibits a protective effect against oxidative stress-induced lung damage in both in vitro and in vivo models. On the basis of these results, the following further studies can be carried out: First, since this study was an acute inflammatory response study, a long-term pre-clinical study might be needed. Further, it is warranted to investigate mechanistic validation(s) with regard to NAR and CPE on PM_10_-induced inflammatory signaling pathways. Finally, clinical trials should be conducted in polluted areas or on smokers who are exposed to PM_10_ in order to determine the clinical potency of CPE and naringin.

## Figures and Tables

**Figure 1 nutrients-14-01101-f001:**
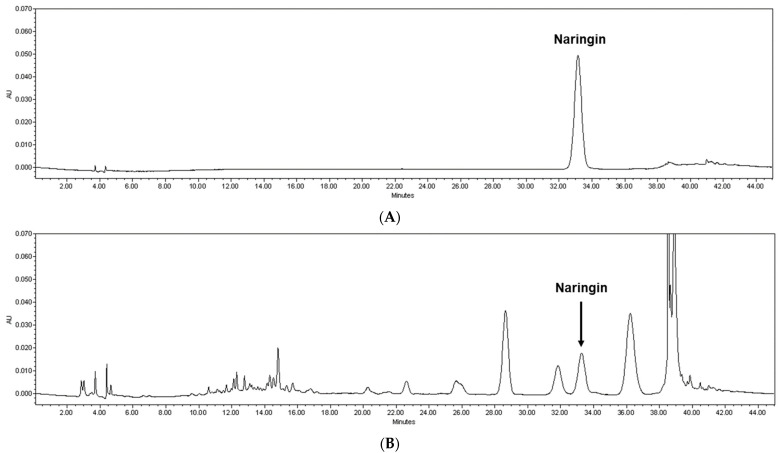
Quantification of naringin from CPE. Representative high-performance liquid chromatography chromatogram (at 280 nm) of the (**A**) naringin reference standard and (**B**) CPE prepared at 60 °C. The arrowed peak has been identified as naringin. CPE, *Citrus junos* Tanaka peel extract.

**Figure 2 nutrients-14-01101-f002:**
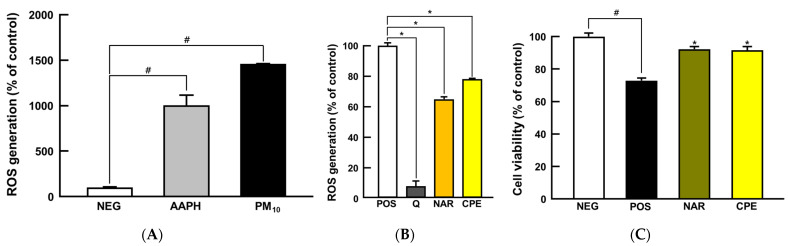
Protective effects of CPE and NAR against PM_10_-induced oxidative stress in NCI-H460 cell line. (**A**) ROS generation, (**B**) intracellular antioxidant activity, and (**C**) cell viability. All values are expressed as mean ± SEM. *p*-value less than 0.05 was considered statistically significant. # indicates statistical significance compared to the NEG group, and * indicates statistical significance compared to the POS group. AAPH, 2,2′-azobis(2-methylpropionamidine) dihydrochloride; CPE, *Citrus junos* Tanaka peel extract; NAR, naringin; PM_10_, particulate matter 10; Q, quercetin; ROS, reactive oxygen species.

**Figure 3 nutrients-14-01101-f003:**
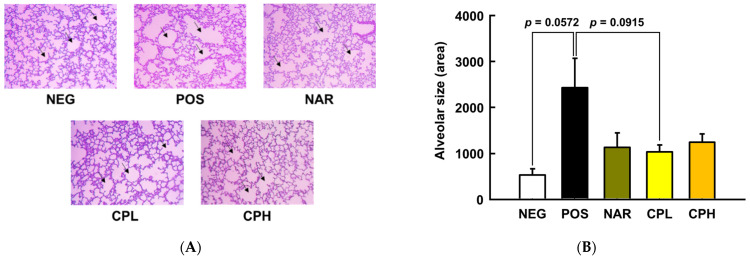
Histological analysis of representative hematoxylin and eosin-stained lung sections, and lung pathology assessment of PM_10_-induced lung damage. (**A**) Hematoxylin and eosin-stained sections; (**B**) alveolar size quantification. All values are expressed as mean ± SEM. CPL, *Citrus junos* Tanaka peel extract low concentration; CPH, *Citrus junos* Tanaka peel extract high concentration; NAR, naringin; NEG, negative control; POS, positive control.

**Figure 4 nutrients-14-01101-f004:**
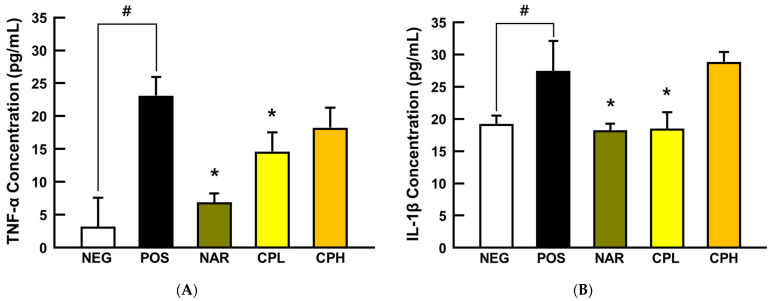
The expression of pro-inflammatory cytokines in the BALF from PM_10_-treated mice. (**A**) TNF-α and (**B**) IL-1β protein expression. All values are expressed as mean ± SEM. *p*-value less than 0.05 was considered statistically significant. # indicates statistical significance compared to the NEG group and * indicates statistical significance compared to the POS group. CPL, *Citrus junos* Tanaka peel extract low concentration; CPH, *Citrus junos* Tanaka peel extract high concentration; NAR, naringin; NEG, negative control; POS, positive control.

**Figure 5 nutrients-14-01101-f005:**
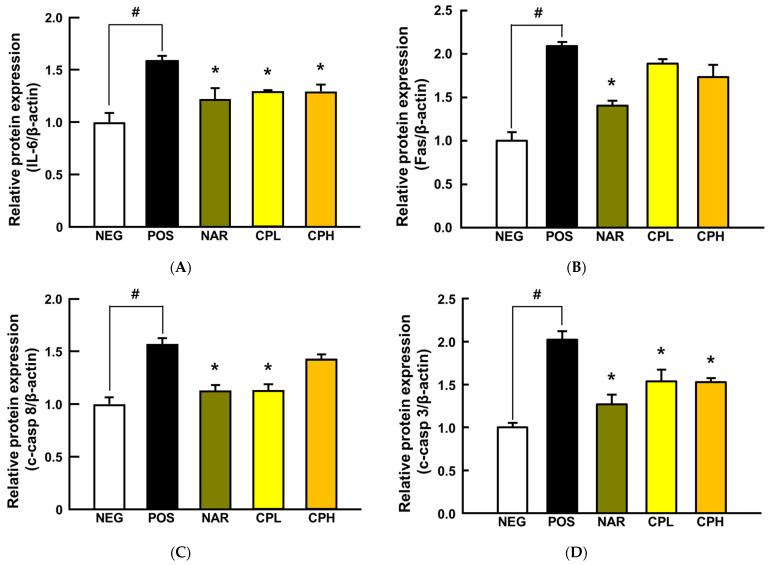
The expressions of pro-inflammatory cytokine- and apoptosis-related proteins in PM_10_-induced lung damage. (**A**) IL-6, (**B**) Fas, (**C**) c-caspase-8, and (**D**) c-caspase-3 expression. All values are expressed as mean ± SEM. *p*-value less than 0.05 was considered statistically significant. # indicates statistical significance compared to the NEG group, and * indicates statistical significance compared to the POS group. CPL, *Citrus junos* Tanaka peel extract low concentration; CPH, *Citrus junos* Tanaka peel extract high concentration; NAR, naringin; NEG, negative control; POS, positive control.

## Data Availability

Not applicable.

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
