# Peer review of "Citrus junos Tanaka Peel Extract and Its Bioactive Naringin Reduce Fine Dust-Induced Respiratory Injury Markers in BALB/c Male Mice"

_nutrients, 2022, doi:10.3390/nu14051101_

Round 1
Reviewer 1 Report
Review Manuscript ID: molecules-1424354
The manuscript entitled "Citrus junos Tanaka peel extract and its bioactive naringin reduce fine dust-induced respiratory injury markers in BALB/c male mice " investigated the role of Citrus Junos Tanaka peel extract (CPE) in preventing the PM10-induced lung injury both in vitro in NCI-H460 cells and an in vivo model of male BALB/c mice. The authors reported that CPE and naringin treatment decreased intracellular ROS production and increased cell viability of lung cells against PM10-induced oxidative stress. Interestingly, they showed the protective effect of CPE against oxidative stress by a PM10-induced lung injury mouse model through histological complementation and suppression of the expression of pro-inflammatory cytokines and extrinsic apoptosis-related proteins.
The introduction is well written, the experimental design is sufficiently complete, and the experiments detailed. The results are well explained, the graphs and the legends are clear. The findings were appropriately discussed in the context of earlier literature. The paper contains new findings and describes how dietary flavonoid naringin and CPE could protect against oxidative stress-induced lung damage through in vitro and in vivo models. The conclusions are consistent with the evidence and arguments presented.
However, some minor revisions are necessary.
1) The authors measured naringin content in CPE by a high-performance liquid chromatography system. I would suggest investigating the complete phenolic profile of CPE, adding more information on the composition of the extract. Such details could help discuss the results obtained in the experiments with naringin alone and with the peel extract.
2) In paragraph 2.4, the authors should specify the dose of naringin or CPE used in the intracellular anti-oxidative assay. Moreover, a dose-effect curve would be needed to identify the best concentration to use in cell treatment.
3) In addition, the choice of CPE concentration used for the treatment of cells and male BALB / c mice and the option of acute administration should be justified either with their own previous studies or with studies in the literature.
Author Response
Response to Reviewer 1 Comments
Point 1: The authors measured naringin content in CPE by a high-performance liquid chromatography system. I would suggest investigating the complete phenolic profile of CPE, adding more information on the composition of the extract. Such details could help discuss the results obtained in the experiments with naringin alone and with the peel extract.
Response 1: We appreciate the reviewer’s thoughtful comments and agree that additional characterization of phenolic compounds profile in CPE would provide helpful information for readers. With all due respect, we do wish to characterize chemical compositions of the CPE yet was not possible as it requires systematic extraction (e.g., liquid-liquid extraction per solvent affinity) followed by multiple LC separations and compound’s structure analyses (e.g., NMR); these separate analytical and chemical analyses would serve as a separate project which is out of scope for the paper. To respond to the reviewer’s comment, however, we executed a literature review about the CPE and potential bioactives present therein (page 4, line 159) and added the discussion about other potential phytonutrients that might have played roles we demonstrated in the study (page 8, line 304).
Point 2: In paragraph 2.4, the authors should specify the dose of naringin or CPE used in the intracellular anti-oxidative assay. Moreover, a dose-effect curve would be needed to identify the best concentration to use in cell treatment.
Response 2: We appreciate the reviewer’s comment. As the reviewer commented, we added the dose of naringin and CPE in paragraph 2.4 (page 3, line 106). Moreover, in order to determine the best concentration, we in fact, executed cytotoxicity tests using the MTT assay (which was not included in the original draft). We, therefore, added the supplementary data in which cytotoxic concentrations of naringin and CPE were determined (Figure S1).
Point 3: In addition, the choice of CPE concentration used for the treatment of cells and male BALB / c mice and the option of acute administration should be justified either with their own previous studies or with studies in the literature.
Response 3: We appreciate the reviewer’s comment on this which is critical. As aforementioned, we determined the in vitro treatment conditions based on the cytotoxicity test (Figure S1). As to the in vivo, prior to our study, we executed a literature review about the concentration of CPE or other natural product extracts in acute pulmonary inflammation models; the related discussion was added to justify the concentration of CPE (page 6, line 208).
Reviewer 2 Report
The aim of this paper was to investigate the effect of Citrus junos Tanaka peel extract against Particulate matter 10 induced lung injury. The manuscript is interesting and well organized. The manuscript fits within the scope of the journal. The title is clear and it is adequate to the content of the article. Only some minor revisions are necessary to improve the clarity of the presentation:
-Please include more information about the plant samples. Include voucher number for plant material.
- Include in the text potential research directions. What are the future applications? What are the next research directions?
Author Response
Response to Reviewer 2 Comments
Point 1: Please include more information about the plant samples. Include voucher number for plant material.
Response 1: We appreciate the reviewer’s comment on this. We do understand that sample identification is critical. As you suggested, we secured an id voucher from an expert and verified our sample information (page 2, line 77).
Point 2: Include in the text potential research directions. What are the future applications? What are the next research directions?
Response 2: The authors thank for the comment. As the reviewer suggested, we revised our conclusion section with a more detailed discussion related to future research directions (page 9, line 330).
Reviewer 3 Report
There are many other ingredients in CPE, authors should prove naringin from CPE is the major protective effect of CPE.
In the animal model, there were two dosages of CPE (100 or 400 mg/kg) treatment groups. The authors should show these two groups' data in the results section. Because the authors only show CPL or CPH on the figures.
Authors tried to present that ROS is a major damage reason from PM10, the authors should add a free radical scavenger as a positive control.
Author Response
Response to Reviewer 3 Comments
Point 1: There are many other ingredients in CPE, authors should prove naringin from CPE is the major protective effect of CPE.
Response 1: We appreciate the reviewer’s thoughtful comments. We do understand that, espeically in natural product extracts, there are numerous bioactives present. Although previous studies have demonstrated that naringin is one of the major active compounds in the citrus family [see below references [1-6]], certainly, there may be other constituents that may elicit protective potential against PM10 either alone or combined. At this moment, however, it is nearly impossible, out of many, to separate impacts of single compound in a biological system while identification/characterization of phytonutrients from the CPE will be a whole separate study. Knowing that we, in fact, described that naringin ‘may explain the protection of CPE at least in part’ (page 8, line 287) which agrees with the reviewer’s prospective. When it comes to justifications in regard to naringin, we believe the reviewer brought a valuable insight hence revised the manuscript in relation to (1) the CPE and potential bioactives present therein (page 4, line 159), and (2) other potential phytonutrients that might have played roles we demonstrated in the study (page 8, line 304).
References
1. Kim, J.W.; Jo, E.H.; Moon, J.E.; Cha, H.; Chang, M.H.; Cho, H.T.; Lee, M.K.; Jung, W.S.; Lee, J.H.; Heo, W. In Vitro and In Vivo Inhibitory Effect of Citrus Junos Tanaka Peel Extract against Oxidative Stress-Induced Apoptotic Death of Lung Cells. Antioxidants 2020, 9, 1231.
2. Xu, G.; Ye, X.; Chen, J.; Liu, D. Effect of heat treatment on the phenolic compounds and antioxidant capacity of citrus peel extract. Journal of Agricultural and Food chemistry 2007, 55, 330-335.
3. Hayat, K.; Zhang, X.; Chen, H.; Xia, S.; Jia, C.; Zhong, F. Liberation and separation of phenolic compounds from citrus mandarin peels by microwave heating and its effect on antioxidant activity. Separation and Purification Technology 2010, 73, 371-376.
4. Shehata, M.G.; Awad, T.S.; Asker, D.; El Sohaimy, S.A.; Abd El-Aziz, N.M.; Youssef, M.M. Antioxidant and antimicrobial activities and UPLC-ESI-MS/MS polyphenolic profile of sweet orange peel extracts. Current research in food science 2021, 4, 326-335.
5. Bocco, A.; Cuvelier, M.-E.; Richard, H.; Berset, C. Antioxidant activity and phenolic composition of citrus peel and seed extracts. Journal of agricultural and food chemistry 1998, 46, 2123-2129.
6. Yu, E.A.; Kim, G.-S.; Jeong, S.W.; Park, S.; Lee, S.J.; Kim, J.H.; Lee, W.S.; Bark, K.-M.; Jin, J.S.; Shin, S.C. Flavonoid profile and biological activity of Korean citrus varieties (II): Pyunkyul (Citrus tangerina Hort. ex Tanaka) and overall contribution of its flavonoids to antioxidant effect. Journal of Functional Foods 2014, 6, 637-642.
Point 2: In the animal model, there were two dosages of CPE (100 or 400 mg/kg) treatment groups. The authors should show these two groups' data in the results section. Because the authors only show CPL or CPH on the figures.
Response 2: We appreciate the reviewer’s comment on this. In fact, we noted that such information was not specified in the materials and methods yet only described in the abstract. The manuscript was now revised as indicated (page 3, line 124).
Point 3: Authors tried to present that ROS is a major damage reason from PM10, the authors should add a free radical scavenger as a positive control.
Response 3: We appreciate the reviewer’s comment. In fact, we used quercetin as a positive control group in cell experiments as quercetin is a powerful free radical. On the other hand, in our in vivo study, we didn’t include a separate positive control group because we have shown that naringin decreased ROS generation in vitro. It is not practically possible to execute a separate animal study to include a positive control group at this moment, yet as we value the reviewer’s comment, this issue was addressed as a potential limitation in the discussion section (page 9, line 308).